# Effects of Fermented Food Consumption on Non-Communicable Diseases

**DOI:** 10.3390/foods12040687

**Published:** 2023-02-04

**Authors:** Priya Patel, Krishna Butani, Akash Kumar, Sudarshan Singh, Bhupendra G. Prajapati

**Affiliations:** 1Department of Pharmaceutical Sciences, Saurashtra University, Rajkot 360005, India; 2Department of Food Technology, SRM University, Sonipat 131029, India; 3Department of Pharmaceutical Sciences, Faculty of Pharmacy, Chiang Mai University, Chiang Mai 50200, Thailand; 4Shree S. K. Patel College of Pharmaceutical Education and Research, Ganpat University, Kherva 384012, India

**Keywords:** fermented foods, gut microflora, intestinal microbiome, non-communicable diseases

## Abstract

The gastrointestinal flora consists of several microbial strains in variable combinations in both healthy and sick humans. To prevent the risk of the onset of disease and perform normal metabolic and physiological functions with improved immunity, a balance between the host and gastrointestinal flora must be maintained. Disruption of the gut microbiota triggered by various factors causes several health problems, which promote the progression of diseases. Probiotics and fermented foods act as carriers of live environmental microbes and play a vital role in maintaining good health. These foods have a positive effect on the consumer by promoting gastrointestinal flora. Recent research suggests that the intestinal microbiome is important in reducing the risk of the onset of various chronic diseases, including cardiac disease, obesity, inflammatory bowel disease, several cancers, and type 2 diabetes. The review provides an updated knowledge base about the scientific literature addressing how fermented foods influence the consumer microbiome and promote good health with prevention of non-communicable diseases. In addition, the review proves that the consumption of fermented foods affects gastrointestinal flora in the short and long term and can be considered an important part of the diet.

## 1. Introduction

Fermentation has long been used to preserve and enhance the shelf life, flavour, texture, and functional properties of food. Humans have been using the fermentation process for thousands of years, mostly to produce alcohol and preserve food. Fermentation is largely an anaerobic process that yields energy for the bacterium or cell while converting carbohydrates to other molecules [1]. Microorganisms with the enzymatic ability for fermentation include bacteria and yeast, specifically the former for lactic acid fermentation and the latter for ethanol fermentation. When these bacteria and yeasts meet the World Health Organization (WHO) criteria of “live microorganisms which, when provided in suitable proportions, impart a health benefit on the host,” they are referred to as “probiotics” [2].

The fermentation process was established long ago. The oldest winemaking techniques were used in Georgia, in the Caucasus region, around 6000 BC, and an alcoholic beverage produced from fruit, rice, and honey dates between 7000 and 6600 BC in the settlement of Jinhu [3]. Roman colonies expanded the practice of making wine throughout the Mediterranean. In Iran’s Zagros Mountains, wine relics were discovered in 7000-year-old jars that are now on display at the University of Pennsylvania. There is strong evidence that alcoholic beverages were being fermented in Babylon in 3000 BC, in ancient Egypt in 3150 BC, and in pre-Hispanic Mexico in 2000 BC and in 1500 BC [4]. Ancient people invented the traditional salting, drying, smoking, and fermenting procedures to preserve food for consumption, marking an important development in the history of human food culture. Based on historical data, various historians have estimated the age of Chinese foods and cuisines to be as old as 4000 BC [4]. Ancient monuments in Nepal show that the region has been consuming Himalayan ethnic meals for more than 2500 years [4].

Foods that have undergone fermentation have a long history. The first time that humanity tasted fermented food may have been a simple accident. The first fermentation must have begun with the preservation of extra milk, as the next day’s produce was fermented. The oldest food preservation technique, before drying, is also known as fermentation. With the advent of civilization, fermentation gained popularity since it not only preserved food but also offered it a diversity of flavours, shapes, and other sensory experiences. Fermented foods have gained popularity as people gradually became aware of their nutritional and medicinal benefits [5].

Foods are fermented using one of two main techniques. Foods can naturally ferment, referred to as “wild fermentation” or “spontaneous fermentation,” which occurs in the presence of microorganisms present in the raw food or processing environment naturally. Examples of such foods include kimchi, sauerkraut, and fermented soy products [6]. The addition of starter cultures, also referred to as “culture-dependent ferments,” to foods, such as natto, kefir, and kombucha, is the second fermentation technique [7].

Fermented foods can be classified into different categories based on (I) the presence or absence of viable microorganisms: (a) fermented foods with viable microorganisms such as non-heated fermented vegetables, kefir, most cheeses, sour cream, miso, yoghurt, tempeh, non-heated salami, natto, pepperoni and other fermented sausages, bushera, boza, and other fermented cereals; (b) fermented foods with no viable microorganisms such as heat-treated or pasteurised fermented vegetables, bread, vinegar, soy sauce, sausage, some kombuchas, distilled spirits, most beers and wine, and chocolate beans (after roasting); (II) classes: (a) cereal products, (b) dairy products, (c) fish products, (d) fruit and vegetable products, (e) legumes, (f) meat products, and (g) beverages; (III) commodity: (a) fermented cereals, (b) alcoholic beverages, (c) fermented vegetable proteins, (d) fermented animal protein and (e) fermented starchy roots; and (IV) commodity (a) cassava-based, (b) cereal, (c) legumes and (d) beverages [8].

Consumers now demand more readily available, nutritious, and safe food. Therefore, it is essential to understand the microbial diversity and nutritional components of traditional fermented foods and their health benefits. Fermented foods are prepared by the controlled and regulated growth of microbes and the enzymatic modification of food components, which serves as a center for microbial consortia [9]. In fermentation, microbes grow naturally or are introduced as starting cultures in the substrates. During fermentation, microbial activity changes the substrates’ biochemical and organoleptic properties [10]. Fermented foods include microorganisms and their metabolites, dietary fibres, and other bioactive components that interact with the intestinal microbiota [9]. Many human intervention studies aim to analyze the effects of fermented food products on the gut microbiota [11,12]. According to recent metagenomic research, fermented foods provide health-promoting microbial species associated with those found in the intestine and may be a significant source of commensal strains [13]. Transient fermented-food-associated microorganisms may combine with intestinal microbiota in the intestine and produce compounds that exhibit immunomodulatory and anti-inflammatory properties [14]. Various studies show that consuming fermented foods may protect against immune- and metabolic-mediated illnesses [15,16,17]. However, to demonstrate the health benefits, fermented foods must have viable health-promoting microorganisms in the range of 10^8^–10^12^ CFU [18,19].

Fermented foods have been linked to several advantages against non-communicable diseases such as diabetes [20], anorexia nervosa [21], weight management [22], cardiovascular diseases [23], irritable bowel syndrome [24], and cancers [25]. During the fermentation process some B-complex vitamins, such as folate, riboflavin, and vitamin B12, are produced from a variety of non-vitamin precursors [26,27]. As a result, the Indian dietary guidelines encourage everyone, particularly pregnant women, to consume fermented foods [10]. Yoghurt is now recommended as part of many nations’ dietary guidelines, including Japan [28], Brazil [29], India [30], Australia [31], and Canada [32]. Fermented foods are excellent delivery systems for microorganisms to the human gastrointestinal tract. Microbes linked to fermentation may change the intestinal flora’s function or composition in the gastrointestinal tract. The extent of these modifications and their significant role in health are still controversial [33,34]. Therefore, there is still a need to ascertain the relationship between fermented foods, altered intestinal flora composition, and host health.

### 1.1. Safety Assessment of Fermented Food for Human Use

The formation of lactic acid, which prevents the majority of viruses from surviving due to the acidic environment present in fermented food and beverages, makes them one of the safer eating options. However, serious processing mistakes that occur in fermented food can potentially put consumer health at risk. Fermented foods may become contaminated with microorganisms that cause food poisoning or spoilage, which would undermine their safety. Additionally, harmful microbes are kept at bay when fermented foods are created using good production techniques and have the right amounts of acid, salt, and sugar. Food safety and nutrition are top priorities in the EU and around the world [35].

Food safety for fermented foods has a long history when they combine low water activity, salt, nitrite, and other antimicrobials with considerable quantities of fermentation-produced organic acids (>100 mM). Likewise, beverages with a pH of less than 4.5 and 4% or more alcohol are regarded as microbiologically safe [36]. By eliminating harmful or anti-nutritional chemicals from the basic components, food fermentation can also improve food safety and nutritional quality. Lactic acid bacteria (LAB) aid in the breakdown of phytate, a substance linked to cereal grains that chelates divalent cations and hinders their absorption in the gastrointestinal tract, during sourdough fermentation [37]. Additionally, it is speculated that sourdough fermentation will lower levels of other immune-reactive proteins, such as the amylase-trypsin inhibitor in wheat [38]. All fermented foods made with filamentous fungi could potentially contain mycotoxins. However, domestication and careful strain selection have successfully eradicated *Aspergillus* and *Penicillium* mycotoxin-producing lineages from cheese, koji, and other fermented foods [39]. Guidelines for the assessment of probiotic safety used in foods were provided by the Food and Agriculture Organization of the United Nations and the World Health Organization [40].

### 1.2. Fermented Food Products Market Potential

During the forecast period, the market for fermented foods and beverages is anticipated to rise at a CAGR of 6.35% (2022–2027). Due to the benefits of health supplements, the COVID-19 pandemic has had an impact on the fermented food and beverage business. However, due to the increased demand for probiotics brought on by the pandemic, fermented food and beverage items have now been given a chance. This tendency is mostly brought on by growing health consciousness and concern over preserving immunity and, by extension, health, and wellbeing.

Foods and drinks that have undergone controlled microbial growth and fermentation are referred to as fermented. In the anaerobic process of fermentation, yeast and bacteria convert food components, such as carbohydrates including glucose, into other compounds (e.g., organic acids, gases, or alcohol). Various probiotic foods, probiotic beverages, alcoholic beverages, and additional categories of fermented food and drink are available. The supermarket/hypermarket, specialty retail store, convenience store, internet channel, and others are the market segments according to distribution channel. The market is divided into five geographic regions: North America, Europe, South America, Asia-Pacific, and the Middle East and Africa. The market size and predictions for each segment have been presented based on the value in USD million [41].

Humans have utilised the fermentation of food from normally perishable raw materials since the Neolithic era (about 10,000 years ago) [42]. Now fast-forward to today, when fermented foods make up one-third of the global human diet [43]. As a result, the usage of food cultures (FC) has considerably risen during the past few decades. Guidelines pertaining to the safety of FC have expanded as a result of this rise. The International Dairy Federation promotes the worldwide dairy industry and makes sure that high-quality milk and wholesome, secure, and sustainable dairy products are supported by the greatest scientific information [42]. Food and Drug Administration (FDA) in the United States and the EFSA in Europe acknowledge that many proposed rules could have a significant economic impact on many small businesses and could impede free and open trade across continents and nations if they are finalised.

Despite being one of the oldest categories of food ingredients in the world, FC is not defined in EU law, and most other countries’ food laws also lack a definition of the term. From 1973 to 2010, Denmark’s national rules inside the EU mandated FC approval. It is still feasible to voluntarily notify and be listed. The European Commission views probiotics as a health claim because no specific strains—aside from the yoghurt microorganisms—have been given authorization for such claims [42,44].

The USA lacks a dedicated FC regulation, much like the EU. For human consumption, some species are deemed “safe and acceptable,” while others are classified as “generally recognised as safe” (GRAS), which is reported to the FDA and published [45]. The legal framework mandates the application of documentation on the strain level for registration to the Food and Drug Administration in order to be included on the list of approved lactic acid bacteria for the fermentation of dairy or beverage products [46]. Foods traditionally fermented (fermented spontaneously) and without the use of FC are considered outside the registration requirements. China has had a special FC legislation since 2010. After 2010, FC that had a history of conventional use could still be used. There is no current list of the traditional FC species used in China. It has been under construction for a few years, and the complete list of species is anticipated to be released soon [42].

The Technical Rule “On Safety of Milk and Dairy Products” (TR TS 033/2013) is the main regulation regulating the standards and requirements for milk and dairy products including the usage of FC [47]. Dairy product producers are responsible for ensuring both the industrial FC’s safety and the manufacturing processes’ compliance with the document’s specifications. The “Bacterial Starters for Dairy Products” basic specifications are currently being developed by the Euro-Asian Council for Standardization, Metrology, and Certification and were anticipated to become official in 2017 [42]. The contemporaneous review of human and animal intervention studies on fermented foods presented relevant scientific findings where the relationship between gut microbiota and non-communicable diseases was examined. In addition, the current review focuses on the overall effect of fermented foods on the gut microbiota rather than the effects of specific types of fermented foods.

## 2. Materials and Methods

Data were retrieved from PubMed/Medline, Scopus, and Google Scholar search engines from October to November 2022, using keywords term such as “fermented foods”, “probiotic”, “gut microbiota”, “health benefits”, “eubiosis” and “fermentation”. The search resulted in several articles, including research papers, reviews, books, patents, and other freely available online sources. Inclusion and exclusion criteria were selected and used to determine recently published articles of importance to produce this document. The publications selected for this study were published on fermented foods and included clinical investigations, review articles, and case reports, all in the English language.

## 3. Effects of Gut Microbiota Interaction with Functional Foods

The greatest microbial community in the human body is found in the gastrointestinal tract. This community, which lives in the small and large intestines, is thought to have more than 100 trillion microbial cells, which is similar to the number of cells in the human body [48]. Previous research has revealed that the gut microflora plays a crucial role in several (patho-)physiological processes, such as the metabolism of specific dietary and pharmaceutical chemicals, the establishment of host immunity, inflammatory conditions in the intestine, and the onset of colon cancer [49]. Throughout the digestive tract, the gut microbiota’s microbial composition fluctuates [50,51]. The gut microbiota is a community of microorganisms composed of 1000–5000 different species of Actinobacteria, Proteobacteria, Firmicutes, and Bacteroidetes [52]. More than 100 trillion microorganisms, or ten times more cells than any other cell in the human body, reside in the intestinal tract. About 30% of the bacteria in the stomach are species from the genus *Bacteroides*, indicating that this genus is particularly crucial to the host’s health [50]. Similarly, there are 3.3 million genes in the human gut metagenome, approximately 150 times as many as in the human genome [53]. The microbial biomass in the human intestine weighs 1.5 to 2.0 kg and is predominately made up of strictly anaerobic bacteria. Microbial density increases from the proximal to the distal end of the intestine [54].

No one undervalues the significance of these enigmatic bacteria, even though at least half of these creatures cannot be grown. In this large population of intestinal flora, anaerobes exceed aerobes by a ratio of 100–1000 anaerobes to one aerobe, according to estimations. However, it is evident that facultative aerobes, such as *Streptococci* and *Escherichia coli*, colonize humans when they are born. During the weaning stage, the gut flora changes dramatically, and obligate anaerobes become dominant (especially *Bacteroides* species) [55]. Intestinal *Bacteroides* that comprise around 30% of the total intestinal flora include at least four major species: *B. thetaiotaomicron, B. vulgatus, B. distasonis, and B. fragilis*, are most likely the main anaerobes in both health and sickness [56]. *Candida, Saccharomyces, Aspergillus, Penicillium, Rhodotorula, Trametes, Pleospora, Sclerotinia, Bullera*, and other fungal taxa have all been found in the gut. Candida is more commonly discovered in people with chronic hepatitis B and cirrhosis of the liver than it is in those with inflammatory bowel disease [57].

In the intestinal tract, the gut microbiota coexists with its hosts in a mutually beneficial way, assisting the host in carrying out a range of biochemical and physiological functions by taking part in various complex metabolic processes as well as the growth and regulation of the immune system. The composition and function of the gut microflora are significantly influenced by nutrition, among other potential factors [58]. The host’s exposure to bioactive food ingredients and their possible health effects can be changed by the gut microbiota’s metabolic activity. Additionally, several food ingredients with functional properties affect the composition and growth of the gut microbiota as well as its metabolic activity [59,60]. Two kinds of “nutraceuticals”, prebiotics and probiotics have drawn the most attention in both fundamental research and product development. Prebiotics are essentially indigestible foods, but through their interaction with the microbes in the gut they are metabolised with modulation in their activity and composition. Prebiotics produce favorable physiological effects on the host as a result of this interaction [61]. Probiotics interact with the gut flora in a similar way as prebiotics. This can occur by producing bacteriocins, or intrinsic “antibiotics,” and antitoxins, competing with pathogens for adhesion, making an environment that is more acidic and hostile to pro-inflammatory bacteria and simultaneously encouraging the growth of beneficial species such as lactobacilli and bifidobacteria [62].

## 4. Role of Different Bacterial and Fungal Species in Fermentation of Edible Foods

The acidity, flavour, and texture of fermented foods, as well as the health advantages that go beyond basic nutrition, all depend on gut microorganisms [63]. Microbes may be present in food as part of its natural microbiota or as a result of their intentional insertion as starter cultures during an industrial food fermentation process [64]. Additionally, microbial cultures can be employed to create a wide variety of substances including enzymes, tastes, and perfumes either intentionally for use as food additives or naturally as a part of food fermentation processes [65]. The extracted microbial enzymes are also used by industries for fermentation, such as α-amylases [66], glucoamylases [66], β-galactosidase [67], α-acetolactate decarboxylase [68] and zymase [69].

During fermentation, bacteria alter the chemical components of raw materials from plant and animal sources, boosting the nutritional content of the goods, enhancing their flavour and texture, extending their shelf life, and bolstering them with bioactive substances that promote health [70]. Asia’s fermented foods and alcoholic beverages are dominated by lactic acid bacteria and Bacillus species, as well as amylolytic and alcohol-producing yeasts and filamentous moulds, whereas Africa is dominated by lactic acid bacteria or a combination of bacterium–yeast combinations and filamentous moulds [71]. Despite the popularity of fermented legume products based on *Bacillus* fermentation in West Africa, filamentous moulds and bacilli are uncommon in fermented foods and beverages in America, Australia, Europe, and Africa [72].

The global supply of food for humans is unavoidably impacted by fermentation, a natural process. Wild fermentation bacteria and yeast are natural resources that are accessible to people all over the world. They are found all over the world in the air, soil, water, and animal guts. They also blanket the continents and penetrate ecosystems [73]. Fermentation is a common occurrence, yet it is poorly understood. Fermentation is the process by which carbohydrates are transformed by microorganisms into substances such as lactic acid and alcohol. Fermented foods frequently introduce microorganisms that live in the human body when consumed by humans [72].

Bacteria: Foods that have undergone spontaneous fermentation as well as those that have been fermented using starter cultures both have bacteria as the predominant microorganism. Lactic acid bacteria (LAB) are more commonly used in the manufacturing of acidic fermented goods than other types of bacteria. Non-LAB bacteria, such as *Propionibacterium, Bifidobacterium, Brevibacterium, Brachybacterium, Micrococcaceae, Bacillus*, etc., are also involved in the fermentation of food, mostly as a secondary group of microorganisms employed to facilitate the efficient progression of the fermentation process [74]. LAB are crucial for food, farming, and medical purposes. Gram-positive, nonsporing, nonrespiring cocci or rods, which create lactic acid as the main byproduct during the fermentation of carbohydrates, are a typical description of the bacteria that make up this group [75]. Certain LAB have been shown to create bacteriocins, which are polypeptides produced by bacteria using their ribosomes and capable of killing or inhibiting the growth of other bacteria [76,77]. Bacteriocins typically cause cell death by preventing the creation of the cell wall or by rupturing the membrane through the formation of pores [78]. Bacteriocins are crucial in food fermentation because they can stop food from spoiling or stop pathogens from growing [79]. LAB create lactic acid and other antimicrobial compounds that prevent the growth of harmful bacteria and lower the amount of sugar in a product, extending its shelf life [71]. Numerous LAB strains have also been found to lower the risk of acute diarrhoea [80]. This research was performed by Rashmi and Sharmila in which the bacteriocin produced by non-lactic acid bacteria from pulses and cow’s milk showed effectiveness against *E.coli*, a Gram-negative, rod-shaped, coliform, facultative anaerobe, that leads to health problems in humans. The bacteria isolated from probiotics showed significant production of antimicrobial substances that can inhibit the growth of food-spoilage bacteria [81]. The term “food-derived components that, in addition to their nutritional worth, exert a physiological effect on the body” refers to bioactive substances. The majority of microbes that ferment lactic acid create organic acids such as lactate, acetate, propionate, and butyrate that have antimicrobial properties. Additionally, LAB are the dominant bacteria in fermented food products that create a variety of high-molecular-mass compounds including anti-microbial peptides and bacteriocins as well as low-molecular-mass molecules such as hydrogen dioxide, carbon dioxide, and diacetyl (2, 3-butanedione) [82].

Non-Yeast Fungi: *Actinomucor, Amylomyces, Aspergillus, Candida, Cladosporium, Penicillium, Pichia, Rhodotorula, Rhodosporidium, Saccharomyces*, and others are examples of different fungal genera. Particularly, filamentous fungi are found in traditional Asian starters with a variety of functions, including liquefaction, saccharification, and ethanol synthesis to create a variety of low- and high-alcohol distilled spirits. They are employed in Europe for the synthesis of enzymes as well as the development of various dairy products and cheese ripening [74]. Citric acid is produced using *Aspergillus* species from waste materials such as apple pomace. Aspergillus species frequently cause food deterioration by inducing unfavorable alterations. On the other hand, cheese ripening and flavour development are related to *Penicillum* species. While *Ceratocystis* species contribute to the creation of fruit flavours, *Penicillium* is the cause of the synthesis of toxins such as patulin [83].

Yeast: In food fermentation, yeasts are advantageous with unfavorable impacts, just like bacteria and other fungi. While some yeasts, such as *Candida*, are used to produce single-cell proteins, others, such as Pichia, are thought to cause food goods to decay. The *Saccharomyces* genus of yeasts, particularly *S. cerevisiae*, which is used in the fermentation of bread and wine to produce alcohol, is best for producing acceptable food fermentation. In the process of creating wine, Saccharomyces cerevisiae var. ellipsoideus is frequently used [84]. Several yeasts, including *Rhodotorula* and *Cryptococcus*, are capable of producing pigment for use as color [85].

The human gut flora has drawn much interest in recent years due to various research findings that it affects both mental and physical health [86,87] and that disruptions to the gut microbiota are linked to several metabolic illnesses [88,89]. Only a few studies have particularly looked at how consuming fermented foods affects the gut flora. Lactic-acid-bacteria-containing fermented foods, such as yoghurt and cultured milk products, improve intestinal health and even cure or prevent a number of disorders [90]. Frequent intake of fermented milk or yoghurt helps in the proliferation of good microbes inside the gut [91,92]. Consumption of fermented foods results in significant increases in beneficial microbes, especially *Bifidobacteria* and *Lactobacilli* [93,94]. Numerous health advantages for consumers are associated with this alteration in the gut ecosystem [95].

## 5. Mechanism of Fermented Foods in Promoting Health

Fermented foods may have positive impacts on health and disease via several processes that contain probiotic microorganisms such as lactic acid bacteria [9]. Most fermented items have been reported to contain at least 10^6^ microbial cells per gram, while quantities can vary based on the region, age, and time of analysis or consumption of the product [7]. By buffering and protecting it against intestinal conditions, the surrounding food matrix appears to be crucial for the survival of probiotic strains. Indeed, numerous studies have demonstrated that bacteria from fermented foods can enter the digestive tract; however, this is likely to vary between products, and their residence in the gut seems to be temporary. Conversely, by competing with pathogenic bacteria and producing immune-regulatory and neurogenic fermentation by-products, these microbes may still be able to exert a physiological advantage in the gut [96].

The modes of action for beneficial fermented food effects include altering the host immunological response, which strengthens resistance to pathogenic challenge, and host microflora at the specific region, including changes in composition and metabolic activity [97]. The gut flora can be improved or restored by different bacteria that are present in fermented foods through a variety of ways involving a large variety of bioactive substances. By altering the pH value of the environment and limiting the oxygen availability, which is critical for the growth of some pathogens, they produce a competitive environment in the gut to suppress the pathogens [98].

Toxins and anti-nutrients can be reduced through fermentation; for instance, soybeans may have lower quantities of phytic acid [99]. Additionally, sourdough fermentation can lower the amount of fermentable carbohydrates (such as fermentable oligosaccharides, disaccharides, monosaccharides, and polyols, or FODMAPs), which may increase patients with functional bowel disorders such as irritable bowel syndrome’s tolerance of these products [38]. The possible and major mechanism of fermented foods are presented in Figure 1.

## 6. Health-Promoting Effects of Fermented Foods

Humans have eaten fermented foods from the dawn of time and utilised them in various forms throughout the world. Fermented milk is the most typical traditional source of *Lactobacilli* [100]. Prebiotics that boost the population of these commensals while eliminating other harmful or neutral species will be selectively fermented by *Bifidobacterium* and Lactobacilli [101,102]. Humans who consume fermented foods have been shown to experience a number of health benefits; these effects have been attributed to the bioactive chemicals that result from microbial fermentation (Table 1).

Consuming traditionally fermented meals prepared from a range of raw materials, microbes, and production methods has gained popularity recently all around the world. Global production of fermented foods and beverages made from milk, vegetables, or fruits is thought to number over 3500 different varieties. Soybean- and cabbage-based fermented foods are a good source of protein, soluble fibre, linolenic and linoleic acid, iron, zinc, vitamin K, vitamin B9, vitamin B1, and vitamin B6. Similar to yoghurt, kefir, and dahi, some fermented milk products also contain calcium, high-biological value (BV) proteins, and vitamins B2, B9, and B12 [106]. Numerous studies have shown that eating foods that have undergone fermentation has health benefits for the host. The consequences of using conventionally fermented foods and drinks to treat the aforementioned conditions are covered in the section that follows.

### 6.1. Fermented Foods against Diabetes

The body experiences persistently elevated blood sugar levels as a result of diabetes or diabetes mellitus. Recently, the potential anti-diabetic properties of a number of fermented dietary components have been investigated [107]. The alcohol content, sugar content, viable cell count, and protein tyrosine phosphatase 1B inhibitory properties of rice wine (Makgeolli) made with various quantities of *Laminaria japonica* have been investigated. The quantity of sugar, alcohol, and microorganisms has not been considerably impacted by the varied *L. japonica* concentrations. Human volunteers accepted the presence of *L. japonica* in Makgeolli at a range of 5–7.5%. The Makgeolli made with 5–7.5% L. japonica shown a high degree of protein tyrosine phosphatase 1B inhibitory action and was generally regarded as acceptable [108]. Newly discovered fermented foods, such as the juice of bitter melon (*Momordica charantia*) when fermented with *Lactiplanti bacillus plantarum* subsp. plantarum, demonstrated anti-diabetic potential in a type 2 diabetic rat model fed a high-fat diet with modest doses of streptozocin. In addition to increasing the concentration of SCFA in the rat model, fermented bitter melon juice was effective in treating hyperinsulinaemia, hyperglycemia, hyperlipidaemia, and oxidative stress [109]. The relationship between insulin resistance, high blood pressure, and fasting hyperglycemia was validated by Irving et al. [110]. Additionally, it was discovered that fasting hyperglycemia and elevated blood pressure together change the dermal microvascular anatomy. In a different study, it was found that giving fructose-induced diabetic rats dahi (an Indian fermented product) containing Lactobacillus acidophilus, Lactobacillus casei, and Lactobacillus lactis reduced the formation of glycogen in the rats’ livers [111]. In another study, red mould and corn silage mould, also known as *Monascus purpureus* Went (Monascaceae) NTU 568, were used to ferment dioscorea roots, long-grain rice, and adlay. The fermented products were fed to diabetes-induced Wistar rats for eight weeks. Rats were evaluated to see if their discomfort from diabetes had improved following the intervention period. The findings demonstrated that the addition of red-mould-fermented products substantially decreased the plasma glucose, triglyceride, amylase, and cholesterol levels, as well as ROS production. The glutathione reductase, glutathione disulfide reductase, and catalase activities all significantly increased in diabetic rats fed red-mould-fermented products [20,112].

### 6.2. Fermented Foods against Cardiovascular Diseases

Worldwide, cardiovascular disease (CVD) is a leading cause of death. Low-density lipoprotein (LDL) cholesterol, a rise in triglyceride-rich lipoproteins, and low levels of high-density lipoprotein (HDL) cholesterol are a few instances of disease-associated hazards that might be either modifiable or unmodifiable factors [113]. Diets including fermented dairy products such as yoghurt and cheese have a favourable or indifferent impact on CVD [114]. Recent research has demonstrated that the probiotics included in fermented dairy foods may have a significant impact on CVD risk by producing certain metabolites that may either directly or indirectly control the development of atherosclerotic plaques [115]. Cowpeas that have undergone natural fermentation have antioxidant and lipid-lowering qualities that may help reduce the risk of cardiovascular disease. In albino Wistar rats fed cowpea-fermented flours for 14 days, significant improvements in plasma antioxidant capacity and hepatic activity of antioxidant enzymes were seen. Plasma cholesterol and triglyceride levels as well as liver weight also improved [116]. Dietary fibre and flavonoid consumption were linked to these benefits [117,118]. Saponins, oligosaccharides, and phytosterols are other minor ingredients that may help to reduce the intestinal absorption of cholesterol. As evidenced by the oral administration of 50% ethanol extracts of red bean natto to Sprague Dawley rats, antioxidants also contribute to the cardioprotective action of fermented legumes [119,120]. Using *Lactobacillus delbrueckii* subsp. Lactis, fermented milk has been shown to have anti-hypertensive benefits by lowering elevated systolic and diastolic blood pressure caused by hypertension [121]. Moreover, consuming cheese was linked to a 2% reduction in CVD risk but had no impact on the chances of CHD or all-cause death. Consuming 50 gram of yoghurt each day had no impact on mortality from any cause, CVD, or CHD risk. This result is unexpected because a 2014 analysis of randomised trials found a link between yoghurt intake and a lower risk of cardiovascular disease (CVD). The authors of the current research speculate that the minimal number of individuals available for this portion of the analysis may have prevented them from demonstrating an association [122].

### 6.3. Fermented Foods against Obesity

Obesity is a condition that is classified as a disease and is characterised by an abnormal or excessive build-up of body fat. Since 1980, the prevalence of obesity and overweight has doubled worldwide, posing a serious threat to public health and placing a heavy burden on individuals and communities. Obesity is the main cause of morbidity and mortality worldwide and a key risk factor for several non-communicable diseases [123]. Obesity and unbalanced gut flora are interlinked with each other. The homeostatic balance of the gut affects downstream metabolites in obese people. For instance, bile acids promote food and vitamin absorption and transport by acting as the primary regulator of lipid metabolism. Bacterial deconjugation and dihydroxylation transform primary bile acid into a secondary form in the small intestine [124]. According to recent research, obesity development may be significantly influenced by unfavourable changes to the gut microbiome’s natural equilibrium [125,126,127]. Therefore, probiotics may help manage obesity by affecting the quantity and nature of the gut microbiota [125,126]. The composition of the metabolites created from dairy products during fermentation frequently relies on the starting culture, the type of milk used, the temperature, and the technological parameters utilised, which may account for some of the variation among studies. Lactic acid bacteria make up the majority of the bacteria in PFMP, although other bacteria or a combination of several types of bacteria, yeasts, and moulds may also be present [128]. Fermented foods are made by bio-transforming the original ingredients through carefully regulated microbial fermentation in a food matrix. Lactic acid, alcohol, acetic acid, propionic acid, bioactive peptides, exopolysaccharides, and other similar chemicals are produced by microbial fermentation in the food matrix [129]. Choi et al. claim that Pueraria Radix fermentation boosted lactate and allowed for the enrichment of specific microbial communities that could help with the anti-obesity process [130]. In addition, research on *C. tricuspidata* modulated *Desulfovibrio, Adlercreutzia, Allobaculum, Coprococcus, Helicobacter, Flexispira,* and *Odoribacter* decreased the levels of alanine aminotransferase, serum triglycerides, and fat mass [131].

### 6.4. Fermented Foods against Cancer

Asian nations account for about half of the world’s cancer cases. According to estimates, 10.6 million more instances of cancer could be diagnosed in 2030 [130]. Urbanization, lifestyle modifications (such as the use of tobacco, alcohol, and an unhealthy diet), ageing, and socioeconomic development will all contribute to an increase in the incidence of cancer. One of the important environmental factors linked to a higher risk of cancer is diet. A healthy diet may lower the risk of developing cancer [132]. Using human acute lymphoblastic leukaemia cells, Horie et al. (2016) examined the anti-cancer properties of *Aspergillus oryzae*-mediated fermented brown rice (FBRE) extract (Jurkat cells). Human acute lymphoblastic leukaemia cells’ viability was decreased by the aqueous extract of FBRE in a concentration- and time-dependent manner. In human acute lymphoblastic leukaemia cells, FBRE increased the expression of death-receptor-related proteins such as membrane-targeted death ligand (truncated Bcl-2 Homology 3 interacting-domain death agonist, tBid), death receptor-5 (DR5), and apoptosis antigen (Fas cell surface death receptor, Fas), while inhibiting the expression of an apoptosis inhibitor (B-cell lymphoma 2, Bc). Trials with inhibitors demonstrated that the caspase-8 inhibitor can reduce cellular toxicity. Collectively, the findings demonstrated that FBRE might cause death-receptor-mediated lymphoblastic leukaemia cell death [133].

The antioxidant content and anticancer properties of naturally fermented beetroot juices derived from beetroots cultivated both organically and conventionally were studied by Kazimierczak et al. (2014). In AGS cells, naturally fermented beetroot juice triggered late apoptosis and necrosis (gastric adenocarcinoma cells). In organically grown beetroot as opposed to conventionally grown beetroot, the bioactivity’s effectiveness was higher. The chemical composition of organic and conventional beetroots may differ, according to metabolomic analysis using ultra-performance liquid chromatography-quadrupole time-of-flight-liquid chromatography-mass spectrometry, and fermented beetroot extracts (extracts made from fermented juice of organic or conventional beetroots) demonstrated a distinguishable chemical property. According to the findings, fermentation of beetroots has a significant impact on the chemical composition and bioactivity, particularly the anti-cancer activity [134].

### 6.5. Fermented Foods against Inflammatory Bowel Disease

In industrialised Asian and Western nations, the incidence and prevalence of inflammatory bowel diseases (IBDs), such as ulcerative colitis and Crohn’s disease, are rising quickly. The majority of patients who stop receiving treatment have a relapse even though biologic medicines that target the immune system have been successful in treating IBD patients, indicating that intrinsic immune dysregulation is an outcome of IBD rather than its primary cause [135]. According to the Brigham and Women’s Health Crohn’s and Colitis Centre, fermented foods such as kefir, kombucha, and sauerkraut contain active probiotics, which are thought of as healthy bacteria and may be beneficial for certain people with digestive issues. A 2018 study that was published in the journal Frontiers of Microbiology in August found that live probiotics in the stomach can also assist in control of the immune system. People with UC may benefit from additional immune support because it is an autoimmune disease. Kombucha tea is an excellent example of a fermented food. Fermented foods are also great as a garnish for any nutritious dish, such as kimchi over rice [136]. For instance, Frank et al. showed that IBD increases the abundance of Actinobacteria and Proteobacteria while decreasing the abundance of bacteria belonging to the phyla Firmicutes and Bacteroidetes [137]. In contrast, Noorbakhsh et al. study’s found that patients who consumed 100 g of yoghurt for four weeks along with lactic acid bacteria (*L. bulgaricus, S. thermophilus, L. plantarum*, and *Limosilactobacillus* fermentum) and a prebiotic (xylooligosaccharides) had significantly less abdominal distention and improved quality of life than the control group [138].

### 6.6. Fermented Foods against Hyperlipidaemia

For lipid-related illnesses including atherosclerosis and coronary heart disease, hyperlipidemia is a frequent risk factor. Genetics and lifestyle are two common reasons for hyperlipidemia. According to research, hyperlipidemia (total cholesterol >200 mg/dL and LDL-C > 100 mg/dL) affected 39.6% of the 19,513 female participants with coronary heart disease [139]. Modulating lipid profiles is more significantly impacted using medical nutrition therapy (dietary regulation and diet modification) [140]. For people with hypercholesterolemia, dietary interventions such as boosting unsaturated fat by consuming *Arbutus unedo* L. [140], lowering saturated fat intake, cholesterol, and increasing fibre-rich foods such as soybeans can decrease total cholesterol [141,142]. Through the binding of bile acids, which promotes the excretion of cholesterol and causes the liver to use cholesterol to produce new bile acids, fibre-rich meals lower blood cholesterol levels. In addition, the bacteria that produce fermented fibre in the gut limit the liver’s ability to make cholesterol. Intestinal bacteria transform the fermented dietary fibre into short-chain fatty acids that can reduce blood cholesterol [139]. According to Kitawaki et al., rats fed soy yoghurt (made with lactic bacteria) had their gene for Elovl 6, a fatty acid elongase that catalyses the conversion of palmitate to stearate, downregulated. Additionally down-regulated was glycerol-3-phosphate acyl-transferase, which produces triacylglycerol and phospholipids. Additionally, consumption of soy yoghurt increased the production of enoyl CoA isomerase, which in turn increased fatty-acid oxidation. The authors came to the conclusion that the regulation of lipid synthesis and breakdown at the gene level was necessary for soy yoghurt to decrease hepatic lipids in these animals [143].

### 6.7. Fermented Foods against Genitourinary Tract Infection

Genitourinary tract infections are a prevalent condition in the immunocompromised host, just like they are in the general population. They can also be separated into genital tract infections and infections of the urinary tract. Genitourinary tract infections are a frequent source of morbidity in both healthy people and people with weakened immune systems. Urinary tract infections (UTIs), which include cystitis, pyelonephritis, and prostatitis, can be broadly divided into two categories: a) UTIs; and b) genital tract infections, which include urethritis, cervicitis, epididymitis, genital ulcerative diseases, endometritis, and pelvic inflammatory disease [144]. In a mouse UTI model, the probiotic *L. casei* strain Shirota was investigated for its ability to inhibit the growth of E. coli. After the challenge, 10^6^ CFU pathogens remained as a chronic infection in the urinary system (bladder and kidneys) for more than 3 weeks. During the period of infection, there was a considerable increase in the number of polymorphonuclear leukocytes and the myeloperoxidase activity in the urine. One dosage of *L. casei* Shirota (10^8^ CFU) given 24 h before to the trial infection effectively suppressed the growth of E. coli and inflammatory reactions in the urinary tract [145].

## 7. Novel Prospects of Fermented Foods

According to the current state of affairs, the post-genomic era of microbiology is now here, and many microbes that are employed in food fermentation or microorganisms that are isolated from food fermentations have already been sequenced. This provides a novel knowledge-based strategy for using microbes for food fermentation, ranging from metabolic engineering of bacteria to create antimicrobials or nutrients to molecular mining of yet-unknown but potentially beneficial actions. A brand-new variety of LAB called fructophilic lactic acid bacteria (FLAB) has been discovered through a recent study; these bacteria prefer fructose to glucose as a growth substrate. In fructose-rich niches, which are the environmental and biological conditions necessary for a species to survive, develop, and reproduce, FLAB are present. P-coumaric acid can be transformed by FLAB enzymes into 4-vinylphenol in the first step and 4-ethylphenol in the second. These secondary metabolites have strong antioxidant properties and are biologically active; they may also improve the flavour of fermented foods [146]. Different fermentation techniques are also developing to improve the fermentation yield and quality of the products. Recent research has shown that using ultrasound during fermentation has an impact on the starting cultures’ activities, producing a distinct fermented product with higher sensory qualities and nutritional quality. The use of low-frequency high-energy ultrasound (LFHEUS), a newly developed technique, has been shown to enhance biological processes such as fermentation [147]. The following are some potential mechanisms for LFHEUS’s beneficial effects on fermentation: increased permeabilization of the membrane, which increases enzyme release, facilitation of mass transfer, and micro mixing; changes in the conformation of the enzymes, which exposes more active sites; and changes in the enzymes’ affinity to substrates are three ways that microbial enzyme activity can be increased [148].

Precision and biomass fermentation to create specific chemicals for the food and chemical sector or medical purposes are recent developments based on genomics and synthetic biology. Biomass fermentation, which relies on microorganisms’ capacity to grow quickly under ideal conditions and create a very high protein content of more than 50% dry weight, is an even more environmentally friendly method of producing protein. Whole-cell biomass produced in this manner can be consumed alone or combined with other dietary components. Marmite manufactured from yeast extract, fermented bean paste, and, more recently, the mycoprotein from a filamentous fungus called Fusarium venenatum, which is utilised as the foundation for meat analogues, are examples of such goods. In order to augment or replace our reliance on animal proteins while both reducing the carbon footprint and boosting nutritional value, biomass fermentation is an appealing alternative [149].

After cereals and grain legumes, tropical root and tuber crops such asyams, colocasia, sweet potatoes, etc. are the third-most significant food crops for humans. They provide either staple or supplementary food for nearly a fifth of the world’s population [150]. These crops play a unique function as a significant staple in several South American, African, and Southeast Asian nations. The human diet is enriched by the large quantities of nutrient-dense, healthy foods that are produced and preserved by fermented foods made from roots and tubers. In Africa, the fermented cassava products include gari, fufu, lafun, chickwanghe, agbelima, attieke, and kivunde; in Asia, tape; and in Latin America, “cheese” bread and “coated peanut.” Cassava is still favoured in Africa and Latin America, and fermentation will likely continue to be a key method of converting this crop into consumables including food, feed, and food additives. Sweet potatoes are a significant crop in China and are used to make a variety of fermented foods, including MSG, lacto-pickles, lacto-juice, sweet potato curd and yoghurt, and edible alcohols [151]. A technology has been developed at the Central Tuber Crops Research Institute (CTCRI), Thiruvanathapuram, India, for the extraction of fermented sweet and sour flour from cassava. In this procedure, cassava roots were fermented using a mixed starting culture. The main class of bacteria and yeasts involved in cassava fermentation are lactic acid bacteria and yeasts [151]. Fermented cassava flour has much higher nutritional qualities and in vitro digestibility than non-fermented flour [152]. Many nations view fermented fish products, such as fish sauce, garam, patis, bakasang, nam-pla, fish paste, and shrimp paste, as basic foods. Many communities across the world receive a large amount of their protein diet from fermented fish products. Fish products that have undergone fermentation are regarded as vital foods in many cultures [153]. Bacteriocins, which work against both Gram-negative and Gram-positive bacteria, are present in several fermented fish products. Lacticin and Weissellicin 110, an organic compound found in Thai fish sauce by the name of plaasom, are present in the Korean fermented fish product known as Jeot-gal, which also resists the activity of Gram-positive bacteria [154]. When fish proteins are fermented, they release bioactive amino acids that have numerous positive effects on human health. Fish bioactive substances, such as fermented cod proteins, are now sold commercially as nutraceuticals that are good for the digestive system [153].

## 8. Conclusions and Future Perspective

Fermented foods have gained popularity recently, largely as a result of their claimed health advantages. Fermented foods became an important part of the diet in many cultures, and over time fermentation has been associated with many health benefits. While fermented foods manufactured in Asia and Africa frequently rely on spontaneous fermentation, those made in Europe, North America, Australia, and New Zealand typically depend on specific starter cultures. Given all these considerations, commercial probiotics must be found to be safe for the general public before being marketed as foods or dietary supplements.

As fermentation can create numerous food types that we generally acquire through a farming-based system, but in a much more sustainable way, it can have a huge impact on the future of our food. Nevertheless, scaling up the production of alternative proteins through fermentation confronts difficulties despite the many advantages. These include production costs, consumer acceptance, and regulatory approval. Additionally, the availability of the genomes of numerous food pathogenic and spoilage bacteria may create new opportunities for the development of novel antibiotics that specifically target these troublesome bacteria’s vital functions. Utilizing this plethora of data for improved cultural performance and activities is the fundamental issue of the genomics and proteomics era as it relates to food systems, enhancing the safety, quality, and composition of our food supply. Although fermentation is one of the oldest human-made technologies, there is still plenty of room for advancement.

## Figures and Tables

**Figure 1 foods-12-00687-f001:**
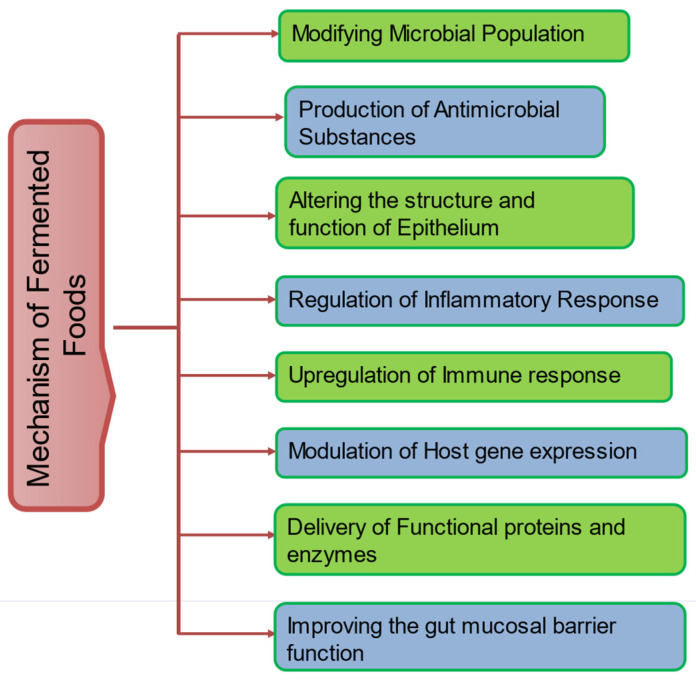
General mechanisms of fermented foods.

**Table 1 foods-12-00687-t001:** Bioactive metabolites found in fermented foods and their activity.

Bioactive	Micro-Organism	Associated Fermented Food	Therapeutic Effect	References
Angiotensin-converting enzyme inhibiting peptides	*Lacticaseibacillus rhamnosus* R0011 and *Lacticaseibacillus helveticus* R0389	Fermented milk	Anti-hypertensive	[103]
Short-chain fatty acids	Various LAB and *Bifidobacterium*	Various fermented food products	Inhibits colorectal cancer	[100]
Kefiran	*Lacticaseibacillus kefiranofaciens*	Kefir	Inhibits cancer	[104]
γ-aminobutyric acid	*Monascus* spp	Red mould rice	Inhibits neurodegenerative diseases	[105]

## Data Availability

The data presented in this study are available on request from the corresponding author.

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
