# Peer review of "Effects of Fermented Food Consumption on Non-Communicable Diseases"

_foods, 2023, doi:10.3390/foods12040687_

Round 1
Reviewer 1 Report
These authors reviewed how fermented foods influence the consumer microbiome and promote good health with the prevention of non-communicable diseases. There are some issues the authors need to address to enhance the suitability of the paper.
General comments
The paper needs proofreading for grammatical errors. Too much "however in sentences".
Quite a lot of sentences in this paper lack appropriate citations e.g. Lines 44, 48, 66, 135, 137, 177, 180, 187, etc.
Re-write Lines 75-81. Confusing!
Should the title of the graphical abstract be different from that of the main title? Please check the format of the Graphical Abstract from the journal.
Change "figure ! in Line 110 to "Figure 1".
Edit Line 113 . Change to "General mechanisms of fermented foods".
Add ";" after "Nutraceutical" in Line 155.
Specific comments
The discussion in the paper is more generic, not specific, and does not add much to existing knowledge.
In Section 5 "The role of different.........". I was expecting to see more discussion on some types of LAB and Nnon-LAB, their functional or beneficial properties, and bioactive compounds added to fermented foods. Also, the title stated ".....bacterial species...", why were fungi and yeast included in the section?
The health-promoting effects discussed in section 6 lack details. There are quite a number of fermented foods but only a few were mentioned.
The novel aspect or prospects of fermented foods such as their prospects as future foods, fermented foods, and alternative proteins, etc., were ignored.
Author Response
Response to Reviewer-1
- The paper needs proofreading for grammatical errors. Too much "however in sentences".
Thank you for the valuable comments
The manuscript have been thoroughly reviewed by native English speaker from Department of English, Ganpat University, Kherva, Gujarat, India
- Quite a lot of sentences in this paper lack appropriate citations e.g. Lines 44, 48, 66, 135, 137, 177, 180, 187, etc.
We, appreciate you for the valuable comments
Appropriate citations added to the relevant text in the draft
- Re-write Lines 75-81. Confusing!
Thank you for the valuable comments
The statement have been paraphrased
Line no.: 75-78; Page no. 2
- Should the title of the graphical abstract be different from that of the main title? Please check the format of the Graphical Abstract from the journal.
Thank you for the valuable comments
Title of Graphical abstract modified
Line no.: 81-82; Page no.: 2
“Impact of fermented food consumption and gut microbiota on non-communicable syndromes”
- Change "figure ! in Line 110 to "Figure 1".
Corrected
Line no.: 370; Page no. 8
- Edit Line 113 . Change to "General mechanisms of fermented foods".
Corrected
Figure 1: General mechanisms of fermented foods
Line no.: 373; Page no. 9
- Add ";" after "Nutraceutical" in Line 155.
Corrected
Line no.: 245; Page no. 6
“nutraceuticals”,
Specific comments
- The discussion in the paper is more generic, not specific, and does not add much to existing knowledge.
The discussions was improved with addition of existing recent reports
- In Section 5 "The role of different.........". I was expecting to see more discussion on some types of LAB and Non-LAB, their functional or beneficial properties, and bioactive compounds added to fermented foods.
Role of LAB and LAB with their functional benefits added in the text
Line no.: 291-314; Page no. 7
Lactic acid bacteria (LAB) are crucial for food, farming, and medical purposes. Gram-positive, non-sporing, non-respiring cocci, or rods, which create lactic acid as the main byproduct during the fermentation of carbohydrates, are a typical description of the bacteria that make up this group [78]. Certain LAB have been shown to create bacteriocins, which are polypeptides produced by bacteria using their ribosomes and capable of killing or inhibiting the growth of other bacteria [79,80]. Bacteriocins typically cause cell death by preventing the creation of the cell wall or by rupturing the membrane through the formation of pores [81]. Bacteriocins are crucial in food fermentations because they can stop food from spoiling or stop pathogens from growing [82]. LAB creates lactic acid and other antimicrobial compounds that prevent the growth of harmful bacteria and lower the amount of sugar in a product, extending its shelf life [74]. Numerous LAB strains have also been found to lower the risk of acute diarrhea [83]. The research was done by Rashmi. D and Sharmila. T. in which the bacteriocin produced from the non-lactic acid bacteria from pulses and cow’s milk shows effectiveness against E. coli, a gram-negative, facultative anaerobe, rod-shaped, coliform which leads to health problems in humans. The bacteria isolated from probiotics showed significant production of antimicrobial substances which can inhibit the growth of the food spoilage bacteria [84]. The term "food-derived components that, in addition to their nutritional worth, exert a physiological effect on the body" refers to bioactive substances. The majority of microbes that ferment lactic acid create organic acids like lactate, acetate, propionate, and butyrate that have antimicrobial properties. Additionally, LABs are the dominating bacteria in fermented food products that create a variety of high-molecular-mass compounds including anti-microbial peptides and bacteriocins as well as low-molecular-mass molecules like hydrogen dioxide, carbon dioxide, diacetyl (2, 3-butanedione) [85].
- Also, the title stated ".....bacterial species...", why were fungi and yeast included in the section?
Thank you for your valuable comments
Heading modified
Line no.: 255-256; Page no. 6
Role of Different Bacterial, Fungal Species, and yeast in Fermentation of Edible Foods
- The health-promoting effects discussed in section 6 lack details. There are quite a number of fermented foods but only a few were mentioned.
Thank you for your valuable comments
The sub-sections for health-promoting effects of fermented food on non-communicable diseases improved with the additions of recent reports
Line no.: 419-427, 441-444, 449-455, 466-469, 481-483, 529-535, 541-545, 562-567; Page no. 10, 11, 12, 13
In another study red mould and corn silage mould, also known as Monascus purpureus Went (Monascaceae) NTU 568, were used to ferment dioscorea roots, long-grain rice, and adlay. The fermented products were fed to diabetes induced Wistar rats for eight weeks. Rats were evaluated to see if their discomfort from diabetes had improved following the intervention period. The findings demonstrated that the addition of red mould fermented products substantially decreased the plasma glucose, triglyceride, amylase, and cholesterol levels, as well as ROS production. The glutathione reductase, glutathione disulfide reductase, and catalase activities all significantly increased in diabetic rats fed red mould fermented products [22,119].
Dietary fibre and flavonoid consumption were linked to these benefits [124,125]. Saponins, oligosaccharides, and phytosterols are other minor ingredients that may help to reduce the intestinal absorption of cholesterol.
Moreover, consuming cheese was linked to a 2% reduction in CVD risk but had no impact on the chances of CHD or all-cause death. Consuming 50 grammes of yoghurt each day had no impact on mortality from any cause, CVD, or CHD risk. This result is unexpected because a 2014 analysis of randomized trials found a link between yoghurt intake and a lower risk of cardiovascular disease (CVD). The authors of the current research speculate that the minimal number of individuals available for this portion of the analysis may have prevented them from demonstrating an association [129].
According to recent research, obesity development may be significantly influenced by unfavorable changes to the gut microbiome's natural equilibrium [132–134]. Therefore, probiotics may help manage obesity by affecting the quantity and nature of the gut microbiota [132,133].
In addition, research on C. tricuspidata modulated Desulfovibrio, Adlercreutzia, Allobaculum, Coprococcus, Helicobacter, Flexispira, and Odoribacter decreased the levels of alanine aminotransferase, serum triglycerides, and fat mass [138].
For instance, Frank et al. showed that IBD increases the abundance of Actinobacteria and Proteobacteria while decreasing the abundance of bacteria belonging to the phyla Firmicutes and Bacteroidetes [144]. In contrast, Noorbakhsh et al study's found that patients who consumed 100 g of yoghurt for four weeks along with lactic acid bacteria (L. bulgaricus, S. thermophilus, L. plantarum, and Limosilactobacillus fermentum) and a prebiotic (xylooligosaccharides) had significantly less abdominal distention and improved quality of life than the control group [145].
Modulating lipid profiles is more significantly impacted using medical nutrition therapy (dietary regulation and diet modification) [147]. For people with hypercholesterolemia, dietary interventions such as boosting unsaturated fat by consuming Arbutus unedo L. [147], lowering saturated fat intake, cholesterol, and increasing fiber-rich foods like soybeans can decrease total cholesterol [148,149].
Genitourinary tract infections are a frequent source of morbidity in both healthy people and people with weakened immune systems. Urinary tract infections (UTIs), which include cystitis, pyelonephritis, and prostatitis, can be broadly divided into two categories: a) UTIs; and b) genital tract infections, which include urethritis, cervicitis, epididymitis, genital ulcerative diseases, endometritis, and pelvic inflammatory disease [151].
- The novel aspect or prospects of fermented foods such as their prospects as future foods, fermented foods, and alternative proteins, etc., were ignored.
Thank you for your valuable comments
Section on Novel Prospects of Fermented Foods
Line no.: 576-641; Page no. 13-14
According to the current state of affairs, the post-genomic era of microbiology is now here, and many microbes that are employed in food fermentation or microorganisms that are isolated from food fermentations have already been sequenced. This provides a novel knowledge-based strategy for using microbes for food fermentation, ranging from metabolic engineering of bacteria to create antimicrobials or nutrients to molecular mining of yet-unknown but potentially beneficial actions. A brand-new variety of LAB called fructophilic lactic acid bacteria (FLAB) has been discovered through recent study; these bacteria prefer fructose to glucose as a growth substrate. In fructose-rich niches, which are the environmental and biological conditions necessary for a species to survive, develop, and reproduce, FLAB is present. P-coumaric acid can be transformed by FLAB enzymes into 4-vinylphenol in the first step and 4-ethylphenol in the second. These secondary metabolites have strong antioxidant properties and are biologically active; they may also improve the flavour of fermented foods [153]. Different fermentation techniques are also developing to improve fermentation yield and quality of the products. Recent research has shown that using ultrasound during fermentation has an impact on the starting cultures' activities, producing a distinct fermented product with higher sensory qualities and nutritional quality. The use of low frequency high energy ultrasound (LFHEUS), a newly developed technique, has been shown to enhance biological processes like fermentation [154]. The following are some potential mechanisms for LFHEUS beneficial effects on fermentation: Increased permeabilization of the membrane, which increases enzyme release, facilitation of mass transfer and micro mixing, changes in the conformation of the enzymes, which expose more active sites, and changes in the enzymes' affinity to substrates are three ways that microbial enzyme activity can be increased [155].
Precision and biomass fermentation to create specific chemicals for the food and chemical sector or medical purposes are recent developments based on genomes and synthetic biology. Biomass fermentation, which relies on microorganisms capacity to grow quickly under ideal conditions and create a very high protein content of more than 50% dry weight, is an even more environmentally friendly method of producing protein. Whole cell biomass produced in this manner can be consumed alone or combined with other dietary components. Marmite manufactured from yeast extract, fermented bean paste, and, more recently, the mycoprotein from a filamentous fungus called Fusarium venenatum, which is utilised as the foundation for meat analogues, are examples of such goods. In order to augment or replace our reliance on animal proteins while both reducing carbon footprint and boosting nutritional value, biomass fermentation is an appealing alternative [156].
After cereals and grain legumes, tropical root, and tuber crops—yams, colocasia, sweet potatoes, etc. are the third most significant food crops for humans. They provide either staple or supplementary food for nearly a fifth of the world's population [157]. These crops play a unique function as a significant staple in several South American, African, and Southeast Asian nations. The human diet is enriched by the large quantities of nutrient-dense, healthy foods that are produced and preserved by fermented foods made from roots and tubers. In Africa, the fermented cassava products include gari, fufu, lafun, chickwanghe, agbelima, attieke, and kivunde; in Asia, tape; and in Latin America, "cheese" bread and "coated peanut." Cassava is still favoured in Africa and Latin America, and fermentation will likely continue to be a key method of converting this crop into consumables including food, feed, and food additives. Sweet potatoes are a significant crop in China and are used to make a variety of fermented foods, including MSG, lacto-pickles, lacto-juice, sweet potato curd and yoghurt, and edible alcohols [158]. A technology has been developed at Central Tuber Crops Research Institute (CTCRI), Thiruvanathapuram, India, for the extraction of fermented sweet and sour flour from cassava. In this procedure, cassava roots were fermented using a mixed starting culture. The main class of bacteria and yeasts involved in cassava fermentation are lactic acid bacteria and yeasts [158]. Fermented cassava flour has much higher nutritional qualities and in vitro digestibility than non-fermented flour [159]. Many nations view fermented fish products, such as fish sauce, garam, patis, bakasang, nam-pla, fish paste, and shrimp paste, as basic foods. Many communities across the world receive a large amount of their protein diet from fermented fish products. Fish products that have undergone fermentation are regarded as vital foods in many cultures [160]. Bacteriocins, which work against both Gram-negative and Gram-positive bacteria, are present in several fermented fish products. Lacticin and Weissellicin 110, an organic compound found in Thai fish sauce by the name of plaasom, are present in the Korean fermented fish product known as Jeot-gal which also resists the activity of Gram-positive bacteria [161]. When fish proteins are fermented, they release bioactive amino acids that have numerous positive effects on human health. Fish bioactive substances, such as fermented cod proteins, are now sold commercially as nutraceuticals that are good for the digestive system [160].
Reviewer 2 Report
The manuscript entitled “Effects of fermented foods consumptions on non-communicable diseases” is very important subject area linking consumption of fermented foods to health and/or specifically to prevention and/or treatment of non-communicable diseases. However, the following needs to be noted for the improvement of the current manuscript.
1. The introduction of the current manuscript comprehensively described the history of fermentation. But the link between fermentation and non-communicable diseases was completely neglected. Moreover, the purpose of this manuscript was left out as well.
2. The general organization and flow of the manuscript itself need to be improved. I would expect the authors to follow the following criteria: Food fermentation - fermented food – consumption of the fermented food - effect of the consumed fermented food on the microbiota – mechanism of fermented food in promoting health – fermented food in prevention and/or treatment of non-communicable diseases. Following this example, then the following suggestions could be proposed.
I. Introduction
II. Role of Different Bacterial Species in Fermentation of Edible Foods
III. Effects of Gut Microbiota Interaction with Functional Foods
IV. Mechanism of Fermented Foods in Exerting Health Beneficial Effects
V. Health Promoting Effects of Fermented Foods
However, some subtitles need to be improved for example “Role of Different Bacterial Species in Fermentation of Edible Foods”. Food fermentation is not only by bacteria as they have included in that subsection. And subtitle 8 to 10 should be part of the introduction.
3. The manuscript needs to be properly cited. For example, “Fermentation has long been used to preserve and enhance the shelf-life, flavor, texture, and functional properties of food. Humans have been using the fermentation process for thousands of years, mostly to produce alcohol and preserve food. Fermentation is largely an anaerobic process that yields energy for the bacterium or cell while converting carbohydrates, like glucose, to other molecules like alcohol. Microorganisms with the enzymatic ability for fermentation include bacteria and yeast, specifically the former for lactic acid fermentation and the latter for ethanol fermentation. When these bacteria and yeasts meet the World Health Organization (WHO) criteria of "live microorganisms which, when provided in suitable proportions, impart a health benefit on the host," referred to as "probiotics” [1].” has only one citation. Moreover, many paragraphs have only one citation just like the example above. Can the authors clarify whether the information from every paragraph come from only one manuscript.
4. The subject matter is not comprehensively described or reviewed. For example, “mechanism of fermented foods in exerting health beneficial effects”, although properly stated and illustrated in figure 1, but how consumption of fermented food affect each of the propose mode of action was not reviewed. And the importance of each mechanism for example, importance of modifying microbial population to the consumer etc. was not reviewed.
5. The health benefits of fermentation foods are a subject area that has taken the world by storm and therefore a lot of literature on how consumption of fermented food affect non-communicable diseases are a lot. But the authors only reviewed on average five to five literature for each disease.
Author Response
Response to Reviewer-2
The manuscript entitled “Effects of fermented foods consumptions on non-communicable diseases” is very important subject area linking consumption of fermented foods to health and/or specifically to prevention and/or treatment of non-communicable diseases. However, the following needs to be noted for the improvement of the current manuscript.
- The introduction of the current manuscript comprehensively described the history of fermentation. But the link between fermentation and non-communicable diseases was completely neglected. Moreover, the purpose of this manuscript was left out as well.
Thank you for the valuable comments
The aim and objective of the review added in appropriate section
Line no.: 191-196; Page no. 5
The review contemporaneous on human and animal intervention studies on fermented foods and presented relevant scientific findings where the relationship between gut microbiota and non-communicable diseases was examined. In addition, the current review focus on the overall effect of fermented foods on the gut microbiota rather than the effects of specific types of fermented foods.
- The general organization and flow of the manuscript itself need to be improved. I would expect the authors to follow the following criteria: Food fermentation - fermented food – consumption of the fermented food - effect of the consumed fermented food on the microbiota – mechanism of fermented food in promoting health – fermented food in prevention and/or treatment of non-communicable diseases. Following this example, then the following suggestions could be proposed.
- Introduction
- Role of Different Bacterial Species in Fermentation of Edible Foods
- Effects of Gut Microbiota Interaction with Functional Foods
- Mechanism of Fermented Foods in Exerting Health Beneficial Effects
- Health Promoting Effects of Fermented Foods
However, some subtitles need to be improved for example “Role of Different Bacterial Species in Fermentation of Edible Foods”. Food fermentation is not only by bacteria as they have included in that subsection. And subtitle 8 to 10 should be part of the introduction.
Thank you for the valuable comments
The manuscript reorganized as suggested with shift of subsection 8-10 in introduction as sub-sections.
- The manuscript needs to be properly cited. For example, “Fermentation has long been used to preserve and enhance the shelf-life, flavor, texture, and functional properties of food. Humans have been using the fermentation process for thousands of years, mostly to produce alcohol and preserve food. Fermentation is largely an anaerobic process that yields energy for the bacterium or cell while converting carbohydrates, like glucose, to other molecules like alcohol. Microorganisms with the enzymatic ability for fermentation include bacteria and yeast, specifically the former for lactic acid fermentation and the latter for ethanol fermentation. When these bacteria and yeasts meet the World Health Organization (WHO) criteria of "live microorganisms which, when provided in suitable proportions, impart a health benefit on the host," referred to as "probiotics” [1].” has only one citation. Moreover, many paragraphs have only one citation just like the example above. Can the authors clarify whether the information from every paragraph come from only one manuscript.
We, appreciate you for the valuable comments
Appropriate citations added to the relevant text in the draft
- The subject matter is not comprehensively described or reviewed. For example, “mechanism of fermented foods in exerting health beneficial effects”, although properly stated and illustrated in figure 1, but how consumption of fermented food affect each of the proposed mode of action was not reviewed. And the importance of each mechanism for example, importance of modifying microbial population to the consumer etc. was not reviewed.
Thank you for your valuable comments
Importance of modifying microbial population to the consumer add in the manuscript
Line no.: 334-343; Page no. 8
The human gut flora has drawn a lot of interest in recent years due to various research findings that it affects both mental and physical health [89,90] and that disruptions to the gut microbiota are linked to several metabolic illnesses [91,92]. Only a few research have particularly looked at how consuming fermented foods affects the gut flora. Lactic acid bacteria-containing fermented foods, such as yogurt and cultured milk products improve intestinal health and even cure or prevent a number of disorders [93]. Frequent intake of fermented milk or yogurt helps in the proliferation of good microbes inside the gut [94,95]. Consumption of fermented foods results in significant increases in beneficial microbes especially Bifidobacteria and Lactobacilli [96,97]. Numerous health advantages for consumers are associated with this alteration in the gut ecosystem [98,99].
- The health benefits of fermentation foods are a subject area that has taken the world by storm and therefore a lot of literature on how consumption of fermented food affect non-communicable diseases are a lot. But the authors only reviewed on average five to five literature for each disease.
Thank you for your valuable comments
The sub-sections for health-promoting effects of fermented food on non-communicable diseases improved with the additions of recent reports
Line no.: 419-427, 441-444, 449-455, 466-469, 481-483, 529-535, 541-545, 562-567; Page no. 10, 11, 12, 13
In another study red mould and corn silage mould, also known as Monascus purpureus Went (Monascaceae) NTU 568, were used to ferment dioscorea roots, long-grain rice, and adlay. The fermented products were fed to diabetes induced Wistar rats for eight weeks. Rats were evaluated to see if their discomfort from diabetes had improved following the intervention period. The findings demonstrated that the addition of red mould fermented products substantially decreased the plasma glucose, triglyceride, amylase, and cholesterol levels, as well as ROS production. The glutathione reductase, glutathione disulfide reductase, and catalase activities all significantly increased in diabetic rats fed red mould fermented products [22,119].
Dietary fibre and flavonoid consumption were linked to these benefits [124,125]. Saponins, oligosaccharides, and phytosterols are other minor ingredients that may help to reduce the intestinal absorption of cholesterol.
Moreover, consuming cheese was linked to a 2% reduction in CVD risk but had no impact on the chances of CHD or all-cause death. Consuming 50 grammes of yoghurt each day had no impact on mortality from any cause, CVD, or CHD risk. This result is unexpected because a 2014 analysis of randomized trials found a link between yoghurt intake and a lower risk of cardiovascular disease (CVD). The authors of the current research speculate that the minimal number of individuals available for this portion of the analysis may have prevented them from demonstrating an association [129].
According to recent research, obesity development may be significantly influenced by unfavorable changes to the gut microbiome's natural equilibrium [132–134]. Therefore, probiotics may help manage obesity by affecting the quantity and nature of the gut microbiota [132,133].
In addition, research on C. tricuspidata modulated Desulfovibrio, Adlercreutzia, Allobaculum, Coprococcus, Helicobacter, Flexispira, and Odoribacter decreased the levels of alanine aminotransferase, serum triglycerides, and fat mass [138].
For instance, Frank et al. showed that IBD increases the abundance of Actinobacteria and Proteobacteria while decreasing the abundance of bacteria belonging to the phyla Firmicutes and Bacteroidetes [144]. In contrast, Noorbakhsh et al study's found that patients who consumed 100 g of yoghurt for four weeks along with lactic acid bacteria (L. bulgaricus, S. thermophilus, L. plantarum, and Limosilactobacillus fermentum) and a prebiotic (xylooligosaccharides) had significantly less abdominal distention and improved quality of life than the control group [145].
Modulating lipid profiles is more significantly impacted using medical nutrition therapy (dietary regulation and diet modification) [147]. For people with hypercholesterolemia, dietary interventions such as boosting unsaturated fat by consuming Arbutus unedo L. [147], lowering saturated fat intake, cholesterol, and increasing fiber-rich foods like soybeans can decrease total cholesterol [148,149].
Genitourinary tract infections are a frequent source of morbidity in both healthy people and people with weakened immune systems. Urinary tract infections (UTIs), which include cystitis, pyelonephritis, and prostatitis, can be broadly divided into two categories: a) UTIs; and b) genital tract infections, which include urethritis, cervicitis, epididymitis, genital ulcerative diseases, endometritis, and pelvic inflammatory disease [151].

Reviewer 3 Report
dear authors,
Although what you are trying to do in this paper is very interesting, the quality of the language is very poor. For example, in line 70 I read of "fermented live microorganisms" and I really wonder what exactly you want to say.
The structure of the paper, the topics, the information you provide are all very interesting but it is a pity that the language does not help the text.
Another issue is that you don't mention the procedure you used to obtain the reference literature. Which key words did you use and by which criteria did you include and exclude papers??
Author Response
Response to Reviewer-3
- Although what you are trying to do in this paper is very interesting, the quality of the language is very poor. For example, in line 70 I read of "fermented live microorganisms" and I really wonder what exactly you want to say.
Thank you for pointing out the corrections
The statement corrected
Line no.: 67-77; Page no.: 2
Fermented foods can be classified into different categories based on (I) the presence or absence of viable microorganisms (a) fermented foods with viable microorganisms such as non-heated fermented vegetables, kefir, most cheeses, sour cream, miso, yogurt, tempeh, non-heated salami, natto, pepperoni, and other fermented sausages, bushera, Boza, and other fermented cereals (b) fermented foods with no viable microorganisms such as heat-treated or pasteurized fermented vegetables, bread, vinegar, soy sauce, sausage, and some kombuchas, distilled spirits, most beers and wine, chocolate beans (after roasting) (II) classes (a) cereal products, (b) dairy products, (c) fish products, (d) fruit and vegetable products, (e) legumes, (f) meat products, and (g) beverages (III) commodity (a) fermented cereals, (b) alcoholic beverages, (c) fermented vegetable proteins, (d) fermented animal protein and (e) fermented starchy roots and (IV) commodity (a) cassava based, (b) cereal, (c) legumes and (d) beverages [9].
- The structure of the paper, the topics, the information you provide are all very interesting but it is a pity that the language does not help the text.
Thank you for the valuable comments
The manuscript have been thoroughly reviewed by native English speaker from Department of English, Ganpat University, Kherva, Gujarat, India
- Another issue is that you don't mention the procedure you used to obtain the reference literature. Which key words did you use and by which criteria did you include and exclude papers??
Thank you for the valuable comments
Detailed methodology added as subsections
Line no.: 67-77; Page no.: 2
Data were retrieved from PubMed/Medline, Scopus, and Google Scholar search engines from October to November 2022, using bullen term such as "fermented foods", "probiotic", "gut microbiota", "health benefits", "eubiosis" and "fermentation". The search resulted in several articles, including research papers, reviews, books, patents, and other freely available online sources, among them considering inclusion and exclusion criteria recently published article of importance were selected and used to produce this document. The publications selected for the study were published on fermented food and included clinical investigations, review articles, and case reports, all in English language.
Round 2
Reviewer 1 Report
The authors have significantly improved the manuscript.
Reviewer 3 Report
I am satisfied with your response to my recomendations